# Dietary Intake Influences Metabolites in Healthy Infants: A Scoping Review

**DOI:** 10.3390/nu12072073

**Published:** 2020-07-13

**Authors:** Mara L. Leimanis Laurens, Chana Kraus-Friedberg, Wreeti Kar, Dominic Sanfilippo, Surender Rajasekaran, Sarah S. Comstock

**Affiliations:** 1Pediatric Critical Care Unit, Helen DeVos Children’s Hospital, 100 Michigan Street NE, Grand Rapids, MI 49503, USA; dominic.sanfilippo@helendevoschildrens.org (D.S.); surender.rajasekaran@spectrumhealth.org (S.R.); 2Department of Pediatrics and Human Development, Michigan State University, Life Sciences Building, 1355 Bogue Street, East Lansing, MI 48824, USA; 3MSU Libraries, 366 W. Circle Drive, East Lansing, MI 48824, USA; krausfri@lib.msu.edu; 4Department of Food Science and Human Nutrition, Michigan State University, Room 139C Trout 469 Wilson Rd, East Lansing, MI 48824, USA; karwreet@msu.edu (W.K.); comsto37@msu.edu (S.S.C.)

**Keywords:** metabolites, infants, human milk, formula milk, serum, plasma, urine, feces

## Abstract

Metabolites are generated from exogenous sources such as diet. This scoping review will summarize nascent metabolite literature and discriminating metabolites for formula vs. human- milk-fed infants. Using the PICOS framework (P—Patient, Problem or Population; I—Intervention; C—Comparison; O—Outcome; S—Study Design) and PRISMA item-reporting protocols, infants less than 12 months old, full-term, and previously healthy were included. Protocol was registered with Open Science Framework (OSF). Publications from 1 January 2009–2019 were selected, for various biofluids, study designs, and techniques (such as high-performance liquid chromatography (HPLC)). From 711 articles, blinded screening of 214 articles using Abstrackr^®^ software, resulted in 24 for final review. Strengthening the Reporting of Observational studies in Epidemiology (STROBE) guidelines were adopted, which included a 24-point checklist. Articles were stratified according to biofluid. Of articles reporting discriminating metabolites between formula- and human milk-fed infants, 62.5% (5/8) of plasma/serum/dried blood spot, 88% (7/8) of urine and 100% (6/6) of feces related articles reported such discriminating metabolites. Overall, no differences were found between analytical approach used (targeted (*n* = 9) vs. un-targeted (*n* = 10)). Current articles are limited by small sample sizes and differing methodological approaches. Of the metabolites reviewed herein, fecal metabolites provided the greatest distinction between diets, which may be indicative of usefulness for future diet metabolite-focused work.

## 1. Introduction

Infant health is strongly correlated to their diet of human milk or infant formula [1,2,3]. Metabolomic analysis of biofluids, such as feces, urine, and blood, is useful to determine the effects of diet on the infant’s development and health [4]. Metabolites are small molecules that are created from diet, chemicals, drugs or tissues when biochemical reactions occur in the body, which influence molecular interactions between genes and proteins leading to a phenotype [5]. Some metabolites are created by microbial processes occurring in the gastrointestinal tract of the individual [6]. The analysis of metabolites may enable individualized medicine allowing for prescribed diets that lead to positive health outcomes [7,8], and metabolomic profiles have been shown to differ between infants fed human milk and those fed infant formula [9].

Both human milk and infant formula affect the infant’s resistance to illness within the first months of life [10,11]. Human milk has several components, such as micronutrients, macronutrients, immune factors, and nutritionally beneficial lipids and oligosaccharides, which support the infant’s immune system [12,13]. Infants fed human milk have fewer gastrointestinal [14,15,16,17] and respiratory illnesses [11,18]. Other benefits of breastfeeding include a reduction of obesity, type 1 and 2 diabetes, leukemia, and allergies [19,20]. On the other hand, contemporary infant formulas are supplemented with key lipids [21,22], a variety of oligosaccharides [23] as well as prebiotics [24,25]. One of the benefits of human milk over formula is that its composition changes in response to both maternal and infant illness [26,27]. A summary of the literature with respect to the metabolites detected in biofluids from infants fed human milk or infant formula with the lens of discriminatory potential has not been explored.

The aim of this scoping review was two-fold: first, to assess the quality of the literature for review, and secondly, to determine whether any of the studies were able to identify metabolites that were discriminatory between breast- and formula-fed infants [28]. The latter aim is important in cases where dietary history is limited, or where parental-recall bias is suspected. In the evolving field of metabolomics, the impact of diet is not well defined. Furthermore, continuously changing formulations of infant formulas to increase their compositional similarity to human milk make the discrimination of infant biofluid from human milk- or formula-fed infants more challenging. This review is intended for general clinicians, nutritionists, dietitians, pediatricians, metabolomics researchers, analytical chemists, and those working in biomarker development, and is unsolicited.

## 2. Materials and Methods 

### 2.1. Protocol Registration

This scoping review was registered at the Open Science Framework database [29] for systematic reviews (https://osf.io/registries), followed the PRISMA reporting (Appendix A), and addressed the following question: “Do formula vs. human-milk-fed infants have different metabolite profiles?” according to PICOS (P—Patient, Problem or Population; I—Intervention; C—Comparison, control or comparator; O—Outcome) (Appendix A).

### 2.2. Search Strategy and Inclusion Criteria

Searches were performed in PubMed, Embase, and Web of Science between 9 May 2019 and 25 June 2019. The search strategy used in PubMed is reported in Appendix A. Searches in Embase and Web of Science were similar and limited to articles published in English, between 1 January 2008 and 31 December 2018. Studies were included if they reported results from healthy, full-term infants less than 12 months of age and reported dietary intake (human milk, formula, mixed diet) as well as metabolomic data from infant biofluid (plasma/serum, stool, urine, dried blood spots (DBS), human milk). All studies, regardless of the technique used to analyze metabolites (such as high-performance liquid chromatography (HPLC), matrix-assisted laser desorption/ionization time-of-flight (MALDI-TOF), nuclear magnetic resonance (NMR)), were included.

Grey literature, opinion papers, letters, case studies, case control studies, and review articles were excluded. Studies focused primarily on older children, premature infants, mothers rather than infants, or non-healthy infants were excluded as well, and studies including both human and non-human infants in the same study were excluded. As cord blood would not reflect human milk or infant formula intake, studies with only cord blood metabolites were also excluded.

### 2.3. Article Screening and Data Abstraction

Authors followed the PRISMA guidelines for item reporting (Figure 1) [30]. All abstracts were screened for inclusion/exclusion by two of the authors using Abstrackr^®^ [31]. After the first round, articles selected for inclusion were then re-screened by the same authors in their full-text form. Disagreements were discussed until a consensus was reached. The initial search yielded 214 relevant results after de-duplication. Of these, 180 were excluded through abstract screening, and a further 10 were excluded during full-text screening (analyzed bacteria in feces rather than metabolites (*n* = 4); analyzed <3 metabolites (*n* = 5); dietary exposure unclear (*n* = 1)). Accordingly, a total of 24 articles were analyzed fully [32,33,34,35,36,37,38,39,40,41,42,43,44,45,46,47,48,49,50,51,52,53,54,55]. This scoping review focused on metabolomic analyses as opposed to individual and single biomarkers, therefore studies with fewer than three metabolites analyzed were excluded from further review.

Abstracted data as summarized in the tables included: the PMID, first author, year of publication, diets, ages, sample sizes (where available initial cohort samples size was listed as well as the sample size for metabolite analysis), assay, targeted vs. un-targeted, whether results were discriminatory, and if so at what age and for which metabolite.

### 2.4. Quality of Reporting Assessment

Given that one of the weaknesses of scoping reviews is the lack of a quality of reporting assessment [56], selected papers were evaluated using the Strengthening the Reporting of Observational Studies in Epidemiology (STROBE) guidelines [57]. These guidelines provide a basis to determine quality of reporting of the studies, providing a minimal list of information to be included for a study to be interpretable and replicable. It is a checklist of items to determine if all the relevant metadata and study design information is reported within the manuscript, as previously described [58]. STROBE guidelines are gaining momentum with journals [59], and have previously been adopted to determine changes in the quality of reporting of the literature, using a pre-/post-test study design [60].

STROBE is designed for observational studies of which 18 (75%) of 24 papers met this criterion (in the case of experimental trials, these were reviewed regardless, and received 1 point for study design in the total checklist). Two reviewers scored each manuscript following STROBE checklist electronically (M.L.L.L., W.K.), and discrepancies were analyzed by a third reviewer (S.S.C). The remaining discrepancies were eliminated through discussions between two of the three reviewers both electronically and verbally (M.L.L.L., S.S.C.), where discrepancies were largely in study design, and sources of bias. Reviewers discussed each case of the remaining discrepancies and came to full agreement. A point was given for each positively evaluated checklist item. From this, the scoring was deemed neutral (16–19) and positive (20–24), no studies were excluded based on their STROBE result.

## 3. Results

### 3.1. Overview

The median age of articles included was 3 years old. International representation of studies included: 7 (29.2%) from Germany, 3 (12.5%) from Japan, 3 (12.5%) from USA, 2 (8.3%) from Italy, 1 (4.2%) from Canada, 1 (4.2%) from Brazil, 1 (4.2%), from Taiwan, 1 (4.2%) from UK, 1 (4.2%) from Poland, 1 (4.2%) from Switzerland, 1 (4.2%) from Netherlands, 1 (4.2%) from Ireland, and 1 (4.2%) from France. In summary, most articles were from Europe (15; 62.5%), 4 (16.7%) from North America, 1 (4.2%) from South America, and 4 (16.7%) from Asia. Study designs included: 8 (33.3%) cohort, 6 (25%) randomized control trials, 6 (25%) case-controls, 2 (8.3%) mother-baby cohort, 2 (8.3%) cross-sectional. The median age of patients recruited was: 4 months. Biofluids included in analyses were: 10 (40.0%) urine, 7 (28.0%) feces, 3 (12.0%) plasma, 3 (12.0%) serum, 2 (8.0%) DBS. Specific measurement techniques included: 6 (24%) liquid chromatography/mass spectrometry (LC/MS); 4 (16%) NMR; 4 (16%) HPLC, 2 (8%) gas chromatography/mass spectrometry (GC/MS), 2 (8%) enzyme-linked immunosorbent assay (ELISA), 2 (8%) matrix-assisted laser desorption/ionization time-of-flight-mass spectrometry (MALDI-TOF-MS), 1 (4%) electrospray ionization-mass spectrometry/mass spectrometry (ESI-MS/MS), 4 (16%) other (1 (4%) high-performance anion-exchange chromatography with pulsed amperometric detection (HPAEC-PAD); 1 (4%) ultra-high-performance liquid chromatography mass spectrophotometry (UHPLC/MS); 1 (4%) chemical analysis; 1 (4%) urease methods).

### 3.2. STROBE

All initial discrepancies were resolved. STROBE results from two independent reviewers revealed total points of neutral (16–19) and positive (20–24) scored manuscripts. All manuscripts met 12 of the 24 STROBE criteria. For seven of the STROBE criteria, fewer than half of the 24 papers reviewed herein met the criteria. These missing criteria included: addressing potential sources of bias, explaining how study size was arrived at, explaining how missing data were addressed, describing any sensitivity analyses, giving reasons for non-participation/sample exclusion at each stage, indicating the number of participants with missing data for each variable of interest, and translating estimates of relative risk into absolute risk for a meaningful time period. Additionally, fewer than 20 studies but more than 12 studies noted the following: how quantitative variables were handled in the analyses, how loss of samples were accounted for, numbers of individuals at each stage of the study, demographic/clinical or social characteristics of study participants, unadjusted estimates of risk, limitations of the study (including sources of potential bias or imprecision), as well as the source of funding and role of the funders in the study. Notably, few articles described how they quantified several of the variables included in their analyses. This directly impacted how the authors described their outcome data. Lastly, the point of generalizability was not always implicitly stated in the text but was assumed to be present when the text presented the greater clinical implications of the work. The weakest sections were: reporting of study size, sample size calculation, and explanation of missing data.

### 3.3. Metabolites

Summary tables were generated for the following biofluids: fecal samples (Table 1), plasma, serum and DBS (included together in Table 2), and urine (Table 3). In cases where manuscripts included both healthy, term-infants, as well as unhealthy and/or pre-term infants, only the healthy infants were discussed, and specific results listed. In manuscripts where more than one biofluid was described, the article is cited in multiple tables. A column was included to describe whether discriminatory markers between diets were found in the outcomes of the studies. This is listed as a separate column in the tables, as well as detailed in the following paragraphs for each biofluid. The use of targeted (*n* = 9) vs. un-targeted (*n* = 10) analytical approach did not reveal differences. Similarly, the techniques used to qualify metabolites was extremely heterogenous, whereby there was little repetition in any of those reporting for discriminatory markers: fecal (*n* = 6 total: GC/MS *n* = 1, LC/MS/MS *n* = 1, H^1^-NMR *n* = 2, MALDI-TOF-MS *n* = 1, HPLC *n* = 1, UHPLC-MS *n* = 1); plasma/serum/DBS (*n* = 5: LC/MS *n* = 2, H^1^-NMR *n* = 1, HPLC *n* = 1, HRMS *n* = 1); urine (*n* = 7: CE-TOF/MS *n* = 2, GC/MS *n* = 1, LC/MS *n* = 1, H^1^-NMR *n* = 1, HPLC *n* = 1, ELISA *n* = 1). From this we may state that H^1^-NMR was more frequently used for fecal metabolite detections, LC-MS for plasma/serum/DBS, and CE-TOF/MS for urine with twice as many times reported. Given the small sample size we are unable to speculate as to which technique is best for each biofluid.

Fecal metabolites were discussed in the following reports [35,37,39,40,44,45] and are summarized in Table 1. Methods for analyzing human fecal metabolites used in these studies employed analytical techniques that were previously validated, and these methods are reviewed in [61]. From review of these studies, age of the infant emerged as an important covariate to measure because metabolomic analyses of stool from infants of various ages are distinct [35,44]. One year of age is a distinct cutoff, where each individual generates a unique repertoire of fecal metabolites, whereas 3–6-month-old infants have similar fecal metabolites if fed the same diet [44]. None of the studies reviewed associated a specific dose of exposure to human milk or formula with discriminatory metabolites. Rather, fecal metabolites, in the reviewed studies, discriminated between any or no exposure to human milk. Discriminatory metabolites included human milk oligosaccharides themselves, which had traversed the intestinal tract intact, and short-chain fatty acids (SCFAs) synthesized by intestinal bacteria. In all studies which included both formula- and human milk-fed infants (*n* = 6 out of 6 such articles), fecal metabolites were discriminatory for dietary exposure.

**Table 2 nutrients-12-02073-t002:** Manuscripts reporting plasma/serum and dried blood spot metabolites from human milk- and/or formula-fed infants (*n* = 8).

PMID	First Author Last Name	Year	Country	Diets	Ages	Sample Size(s)	Assay	Targeted or Untargeted Analysis	Discriminatory Metabolites for Exclusive Human Milk Diet
28190990	Acharjee	2017	United Kingdom	Exclusive human milk; exclusive formula; mixed diet	3 and 6 mos of age	*n* = 374; *n* = 416 samples	HRMS	Untargeted	Yes, *DBS* revealed PC(35:2), SM(36:2), and SM(39:1), these markers can be used to determine whether infants were exclusively FF or human milk-fed
29856767	Hellmuth	2018	Germany	Exclusive human milk	4 mos	*n* = 137	LC-MS/MS	Targeted	No, *serum* phospholipids, acylcarnitines, and amino acids characterized; LPC 14:0, associated with obesity risk and associated to human milk protein content
25368978	Kirchberg	2015	Germany	Exclusive human milk; exclusive formula (HP vs. LP)	6 mos	*n* = 691; *n* = 764 samples	LC-MS	Targeted	Yes, 29 *plasma* metabolites differed, BCAA’s were the most discriminant.
22488223	Neto	2012	Brazil	Majority exclusively human milk fed infants	4–8 days	*n* = 106	ESI-MS/MS	Targeted	No, *DBS* from heel prick blood samples, of majority exclusively human milk-fed infants. Long-chain acylcarnitines, palmitoylcarnitine, stearoylcarnitine, and oleolycarnitine increased by 27%, 12% and 109% in the first week of life.
28623320	Slupsky	2017	USA	Exclusive human milk; exclusive formula (lactose vs. CCS)	3 mos (± 2 weeks)	*n* = 34; *n* = 30 samples	^1^H-NMR	Untargeted	Yes, FF infants demonstrated a rapid increase in circulating *plasma* amino acids, creatinine and urea compared with human milk-fed infants within 2 h postprandial.
21849603	Socha	2011	Poland	Exclusive human milk; exclusive formula (HP vs. LP)	6 mos	*n* = 764	HPLC	Targeted	Yes, most essential *serum* amino acids, IGF-1, and urea increased significantly in both the LP and HP groups compared to human milk.
27571269	Uhl	2016	Germany	Exclusive human milk; exclusive formula (IF-AA and DHA vs. CF)	0–4 mos	*n* = 231; *n* = 484 plasma samples	LC-MS	Targeted	Yes, CF group showed 40% (AA) and 51% (DHA) *plasma* levels as compared to human milk-fed infants.
22100021	Wu	2011	Taiwan	Exclusive human milk; exclusive formula	1 and 2 mos olds	*n* = 60; *n* = 120 samples	Chemical analysis	Targeted	No; *Serum* cholesterol, TG, ALT, AST, GGT, T-bil and D-bil levels were significantly higher in the human milk-fed group, at both 1 and 2 mos; BUN and IP levels were significantly lower in the human milk-fed group compared with the FF group.

Abbreviations: ALT: alanine aminotransferase; AST: aspartataminotransferase; BCAA’s: branched-chain amino acids; BUN: blood urea nitrogen; CCS: corn-syrup solids; CF: control formula; D-bil: direct bilirubin; DBS: dried-blood spots; DCA, dicarboxylic acid; ELISA: enzyme-linked immunosorbent assay; ESI-electrospray ionization; FF: formula-fed; GC/MS: gas chromatography; GGT: glutamyl transferase; HMO, human milk oligosaccharides; HP: high-protein; HPLC: high-performance liquid chromatography; HRMS: high-resolution mass spectrometry; IF: isoenergenic formulae (intervention formula with equal amounts of arachidonic acid (AA) and docosahexaenoic acid (DHA)); IGF-1: insulin-like growth factor-I; IP: inorganic phosphate; LC: Liquid chromatography; LP: low-protein; LPC: lyso-phosphatidylcholine; MALDI-TOF: Matrix-Assisted Laser Desorption/Ionization-Time Of Flight; mos: months; MS: mass spectrometry; NMR: nuclear magnetic resonance; PC: phosphatidylcholine; SM: Sphingomyelin; T-bil: total bilirubin; t-PGDM: tetranor prostaglandin D2 metabolite; TOF: time-of-flight; TG: triglyceride.Urine sample outputs were reported in a total of 10 of the studies [34,38,40,42,44,49,50,51,55,62] and the outcomes and trends are presented in Table 3. Scalabre et al. was finally excluded from further review due to a lack of explicit detailing in methods on dietary intake [47].

**Table 3 nutrients-12-02073-t003:** Manuscripts reporting urine metabolites from human milk- and/or formula-fed infants (*n* = 10).

PMID	First Author Last Name	Year	Country	Diets	Ages	Sample Size(s)	Assay	Targeted or Untargeted Analysis	Discriminatory Metabolites for Exclusive Human Milk Diet
27153855	Anderson	2016	USA	Exclusive human milk; exclusive formula	Mean age of 3.09	*n* = 175	Urease method	Targeted	No; DCA (adipic, suberic, sebacic acids) excretion amounts did not differ between groups.
27650928	Cesare Marincola	2016	Italy	Exclusive human milk; exclusive formula (functional vs standard)	Enrollment (T0); 60 days (T1); 130 days (T2)	*n* = 60; *n* = 120 samples	^1^H-NMR	Untargeted	Yes, Age-dependent differences for choline, betaine, myoinositol, taurine, and citrate for 3 types of nutrition; no differences between two formulas.
26907266	Dessi	2016	Italy	Exclusive human milk; exclusive formula	At birth (T0) (prior to food); Day 3 (T1) of life, Day 7 of life (T2)	*n* = 14; *n* = 31 urine samples	GC-MS	Untargeted	Yes, At 3 days formula milk higher levels of glucose, galactose, glycine and myo-inositol; aconitic, aminomalonic, adipic acids elevated in human milk-fed.
25330044	Dotz ^a^	2015	Germany	Exclusive human milk	2–6 mos	*n* = 10; 76 urine samples	MALDI-TOF-MS (/MS)	Targeted	*n*/A; novel human HMO detected; see also table of fecal metabolites.
28095889	Hill	2017	Ireland	Exclusive human milk	1–24 weeks	*n* = 192	LC-MS	Untargeted	*n*/A; pre-term vs. full-term collected at 4 weeks demonstrate a functionally different metabolite profile.
24375085	Martin	2014	Switzerland	Exclusive human milk; exclusive formula	3, 6, 12 mos	*n* = 236; 587 urine samples	ELISA	Untargeted	Yes, FF infants different from breast-fed, lipid and energy metabolism (carnitines, ketone bodies, and Krebs cycle).
									see also table of fecal metabolites
21849603	Socha	2011	Poland	Exclusive human milk; exclusive formula (HP vs. LP)	6 mos	*n* = 636	HPLC	Targeted	Yes, urinary C-peptide: creatinine ratio higher in HP, as compared to LP and human milk-fed.
28777736	Shoji ^b^	2017	Japan	Exclusive human milk; exclusive formula	1 and 6 mos	1 mth (*n* = 19); 6 mos (*n* = 14)	CE-TOF/MS	Untargeted	Yes, Choline metabolites (choline base solution, *n*,*n*-dimethylglycine, sarcosine, and betaine) and l(-)-threonine and l-carnosine excretion at 1 mos were statistically significantly higher in human milk-fed infants; 1(-)-threonine and 1-carnosine. Lactic acid was lower in human milk-fed infants at both 1 and 6 mos.
28898456	Shoji ^c^	2018	Japan	Exclusive human milk; exclusive formula	1 and 6 mos	1 month (*n* = 13 human milk; *n* = 6 formula); 6 mos (*n* = 9 human milk; *n* = 5 formula)	LC-MS	Targeted	Yes, Urinary t-PGDM at 1 and 6 mos was significantly lower in breastfed infants than FF.
29886808	Shoji ^d^	2018	Japan	Exclusive human milk; exclusive formula	Day 15–1 mos	*n* = 39	CE-TOF/MS	Targeted	Yes, Choline metabolites (choline, *n*,*n*-dimethylglycine, sarcosine, and betaine) differed between human milk-fed and term FF.

Abbreviations: CE: capillary electrophoresis; DCA, dicarboxylic acid; ELISA: enzyme-linked immunosorbent assay; ESI-electrospray ionization; GC/MS: gas chromatography; HMO, human milk oligosaccharides; LC: Liquid chromatography; MALDI-TOF: Matrix-Assisted Laser Desorption/Ionization-Time Of Flight; MS: mass spectrometry; NMR: nuclear magnetic resonance; t-PGDM: tetranor prostaglandin D2 metabolite; TOF: time-of-flight; mos, months; FF, formula-fed; Dotz ^a^: [40]; Shoji ^b^: [55]; Shoji ^c^: [49]; Shoji ^d^: [50]; N/A: no comparative group listed.

The discriminatory ability of the articles was qualified as either yes (*n* = 5), no (*n* = 3), and largely based on the detection of lipids and amino acids. In summary, Acharjee et al. found that an exclusively human milk diet resulted in the presence of distinct lipids species (PC (35:2), SM (36:2), SM (39:1) [32]. Kirchberg determined that a total of 29 plasma metabolites differed between dietary groups, of which BCAA’s were the most discriminant [43]. Slupsky et al. noted increased circulating plasma amino acids, creatinine and urea compared with human milk-fed infants within 2 h postprandial [48]. A similar observation was made with Socha et al. where essential serum amino acids, IGF-1, and urea increased significantly in both the LP and HP groups compared to human milk [51]. Finally, Uhl et al., control formula showed 40% (AA) and 51% (DHA) plasma levels as compared to human milk-fed infants [52].

In spite of the lack of a comparative group, other observations in this sub-cohort of studies looking at plasma, serum and DBS, from exclusively human milk-fed included those describing serum phospholipids, acylcarnitines, amino acids, and LPC 14:0 [41]. This finding was similar to that of Neto et al. which found long-chain acylcarnitines, palmitoylcarnitine, stearoylcarnitine, and oleolycarnitine increased by 27%, 12% and 109% in the first week of life, this article however, did not evaluate any additional time points [46]. Additional markers for the exclusively human milk-fed included higher levels of serum cholesterol, TG, ALT, AST, GGT, T-bil and D-bil levels were significantly higher in the human milk-fed group, at both 1 and 2 months [53].

Dietary-specific urinary markers were found in seven studies [38,44,49,50,51,55,62]. Cesare Marincola et al. found age-dependent differences for choline, betaine, myoinositol, taurine, and citrate for three types of nutrition, and no differences between the two formula types [62]. In Dessi et al. it was determined at three days that infants fed formula milk had higher levels of glucose, galactose, glycine and myo-inositol; aconitic, aminomalonic, adipic acids were elevated in human milk-fed infants [38]. Martin et al. was listed in both the table for fecal metabolites and urine, which found that at 3-, 6-, and 12-month, formula-fed differed from breast-fed infants, based on both lipid profiles and energy metabolism (carnitines, ketone bodies, and Krebs cycle); this study was powered with a large sample size (*n* = 236) and many urine samples (*n* = 587) [44]. Socha et al. looked at infants (*n* = 636) at six months, where urinary C-peptide: creatinine ratio higher in HP, as compared to LP compared to human milk-fed [51]. Shoji et al. (2017) found that choline metabolites (choline base solution, *N*, N-dimethylglycine, sarcosine, and betaine) and l(-)-threonine and l-carnosine excretion at 1 month were statistically significantly higher in human milk-fed infants; 1(-)-threonine and 1-carnosine. Lactic acid was lower in human milk-fed infants at both 1 and 6 months [55]. Another study by the same group reported urinary t-PGDM at 1 and 6 mos was significantly lower in breastfed infants than FF [49]. A final study by the same group found that choline metabolites (choline, N, N-dimethylglycine, sarcosine, and betaine) differed between human milk- and term formula-fed [50].

One manuscript did not provide discriminatory results. Anderson et al. looked at exclusive human vs. exclusive formula milk in 175 infants and DCA (adipic, suberic, sebacic acids) excretion amounts did not differ between groups [34]. Two additional manuscripts did not have a comparative group [40,42].

## 4. Discussion

The current review of the literature describing metabolites in biofluids of healthy infants fed human milk, infant formula or a mixed diet demonstrates wide variability in metabolites but provides evidence that metabolites in fecal samples may be best suited to differentiating human milk-fed from formula-fed infants. Furthermore, it highlights the variety of methods used to profile these metabolites and the failure of much of the literature to report necessary details about study design, participants, and generalizability of reported results. It was beyond the scope of this review to determine if compositional differences within human milks or within formulas can be detected in biofluids from infants consuming those foods. However, one of the manuscripts that was reviewed compared formulas of differing composition (high- and low-protein formulations). That article did not identify differences in infant metabolites by formula composition. Another of the manuscripts (Dotz 2015 [40]) did identify fecal metabolite differences related to differences in the human milk oligosaccharide composition of the milk.

Based on the STROBE results and to report on the primary objective of the work, we can deduce that the overall quality of the articles is fair to positive. None of the articles were excluded based on their STROBE results and this internal metric should be considered for future reviews, as it is not customary for scoping reviews to screen articles using this metric.

This literature review also demonstrates that metabolites are measured using a variety of methods, including targeted (by MS only) or untargeted (by NMR or MS) approaches. Targeted approaches are those which set out to measure specific metabolites whereas untargeted approaches measure all metabolites [63]. Typically, targeted approaches allow for quantitative assessment of metabolite abundance whereas untargeted can only compare relative amounts within the study population. Untargeted approaches are often used because of their ability for discovery of new metabolites. Variability in methodologies was not differentiated, given the limited number of manuscripts using each technique that were included in the final analysis.

Blood, plasma, and DBS are attractive sources of metabolic differences. It is known that blood reveals metabolic changes reflective of organ function and is commonly used in clinical biochemistry, therefore there is ongoing interest to develop blood markers of interest for this patient population. The use of blood for patient analytics remains at the forefront of current bedside medicine, playing a major role in decision making, deductive reasoning for physicians and how we derive a diagnosis. It has evolved in the field as a common biofluid of interest [5].

There also is a desire in the pediatric community to determine novel ways to run studies using non-invasive practices, of which urine and fecal matter, both constituted as waste-products, have a certain appeal. However, urine has specific challenges [64,65,66]. For instance, urine must be considered over a 24-h period of time, as opposed to a single spot urine, for some analytes [67,68]; however, this can be impractical in infant populations [68]. Fecal samples may be an attractive specimen for collection, but this sample type requires that the patient be consuming a diet enterally. In the case of hospitalized patients, they often are not consuming an enteral diet and may be *nil per os* (NPO). In such cases, little fecal matter is produced, making it difficult to use this biofluid for metabolite analyses. Nonetheless, 100% (6/6) of articles describing fecal metabolites reported discriminatory metabolites between formula- and human milk-fed infants. These discriminatory metabolites included the human milk oligosaccharides themselves as well as short-chain fatty acids. Only 62.5% (5/8) of plasma/serum/dried blood spot and 88% (7/8) of urine reports had such discriminating metabolites. Thus, when the goal is to discriminate between human milk- and formula-fed infants, feces may be the preferred biospecimen.

### Limitations

This review excludes articles published after 31 December 2018. Findings cannot be used to recommend policy/practice. In many cases biomarkers still need to be validated in a separate cohort. Given the limited number of manuscripts, we were not able to further stratify based on technique, which may introduce additional heterogeneity. Studies may not have been adequately powered. We did not differentiate here between human milk-fed to infants at the breast or human milk given to infants via bottle feeding.

## 5. Conclusions

There are many articles that characterize dietary metabolites in infants, as described here (*n* = 24). Most are of high quality, but some do not clearly describe the populations from which samples were collected nor the methods used to handle missing data/samples. These articles reported differentiating between human milk- and formula-fed infants based on metabolites in their urine, plasma, serum, blood or feces. Although the studies identified several metabolites that differ between human milk- and formula-fed infants, only some metabolites were reported in more than one study. These are summarized here. Human milk oligosaccharides and short-chain fatty acids were discriminatory in fecal samples. In serum/plasma, amino acids and urea were discriminatory. For urine, myoinositol was discriminatory. It is beyond the scope of this review to state whether sufficient evidence exists; however, if dietary markers are of interest, then the studies listed in this review may serve as reference body of literature. Further work is required. In healthy infants, fecal samples provide the best discriminatory prospect for dietary metabolite differentiation.

## Figures and Tables

**Figure 1 nutrients-12-02073-f001:**
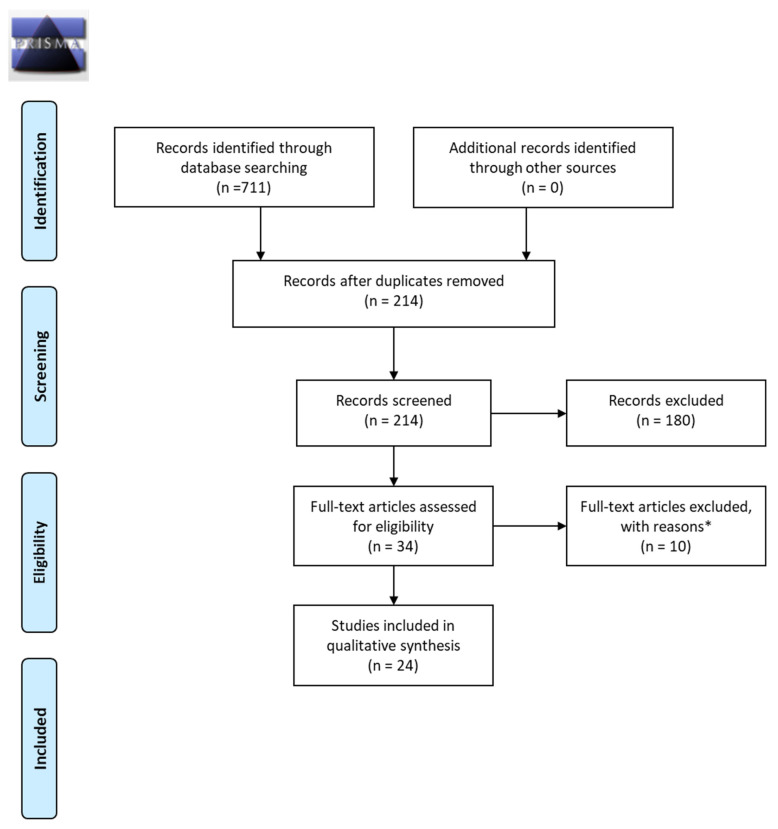
PRISMA 2009 Flow Diagram. * Note: Full-Text articles excluded with reasons (*n* = 10): analyzed bacteria in feces rather than metabolites (*n* = 4); analyzed single or <3 metabolites (*n* = 5); dietary exposure unclear (*n* = 1).

**Table 1 nutrients-12-02073-t001:** Manuscripts reporting fecal metabolites from human milk- and/or formula-fed infants (*n* = 7).

PMID	First Author Last Name	Year	Country	Diets	Ages	Sample Size(s)	Assay	Targeted or Untargeted Analysis	Discriminatory Metabolites for Exclusive Human Milk Diet
24628373	Chow	2014	USA	Exclusive human milk; exclusive formula	Fed diet for at least 2 mos (likely between 2–6 mos)	*n* = 4 per diet	GC/MS; LC/MS/MS	Untargeted (>250 metabolites)	Yes, HMO and their metabolites; fewer protein fermentation metabolites
24375085	Martin ^a^	2014	Switzerland	Exclusive human milk; exclusive formula; until 6 mos	3, 6, 12 mos	*n* = 111	^1^H-NMR	Untargeted	Yes, 1 year unique from 3 and 6 mos; 3 and 6 similar; human milk-fed had high concentrations of fucosylated oligosaccharides and lactic acid; FF characteristic SCFA profile, higher propionate, butyrate, acetate, 5-amino-valerate at 3 and 6 mos. FF higher free amino acids (phenylalanine, tyrosine, leucine, and isoleucine).
25330044	Dotz ^b^	2015	Germany	Exclusive human milk	2–6 mos	*n* = 6	MALDI-TOF MS; HPAEC-PAD	Targeted: HMO and HMO metabolites	*n*/A; Infants separated into 3 groups of HMO metabolites
27613801	Dotz ^c^	2016	Germany	Exclusive human milk; exclusive formula; mixed-fed	2 mos and 7 mos	*n* = 23 exclusive human milk; *n* = 11 exclusively formula, *n* = 1 mixed	MALDI-TOF-MS;	Targeted: HMO and HMO metabolites	Yes, the few signals detected in FF infants were at very low intensity.
27362264	Martin ^d^	2016	Netherlands	Exclusive human milk; exclusive formula; mixed-fed	1st stool, 2 days post-first stool, 1 week, 1, 3, and 6 mos, 1 week post-weaning	*n* = 108	HPLC	Targeted: SCFAs	Yes, human milk-fed had higher proportion of lactate and lower proportion of butyrate. Solid food led to lower proportion of succinate and lactate with more butyrate.
28877893	Bazanella	2017	Germany	Exclusive human milk; exclusive formula	Monthly 1st year of life, again at age 2 years	*n* = 106 (9 of these were exclusively human milk-fed); *n* = 70 at 2 years	UHPLC/MS	Untargeted and Targeted: SCFAs, HMO	Yes, human milk-fed differed from FF in sterol lipids, glycerophospholipids, fatty acids. A main discriminating human milk metabolite (*m/z* 552.3366) was a glycerophospholipid. A main discriminating FF metabolite (*m/z* 407.2455) was a sterol lipid-like molecule. Human milk-fed had lower proportions of propionate, butyrate, valerate and isovalerate but higher proportions of pyruvate and lactate. Propionate, butyrate, isovalerate, valerate increased with time in all.
28443284	Bridgman	2017	Canada	Exclusive human milk; exclusive formula; mixed-fed	3–5mos (mean age 3.7 ± 0.5 mos)	*n* = 163 (30% only human milk, 28% only formula, 42% mixed) CHILD	^1^H-NMR	Untargeted	Yes, human milk-fed had higher proportion of acetate than FF.

Abbreviations: CS, chondroitin sulfate; DS, dermatan sulfate; FF, formula-fed; GC, gas chromatography; HS, heparin sulfate; HA, hyaluronic acid; HMO, human milk oligosaccharides; HPAEC-PAD, high-performance anion-exchange chromatography with pulsed amperometric detection; HPCE, high performance capillary electrophoresis; HPLC, high performance liquid chromatography; LC, liquid chromatography; MALDI, matrix-assisted laser desorption/ionization; mos, months; MS, mass spectrophotometry; NMR, nuclear magnetic resonance; SCFA, short-chain fatty acids; TOF, time-of-flight; UHPLC/MS, ultra-high performance liquid chromatography mass spectrophotometry; Martin ^a^: [44]; Dotz ^b^: [40]; Dotz ^c^: [39]; Martin ^d^: [45]; *n*/A: no comparative dietary group listed. The summary of plasma, serum and DBS are included in Table 2. Plasma and serum were the biofluids for the following reports [41,43,48,51,52,53]. Two articles discussed metabolites as revealed from dried-blood spot samples [32,46].

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
