# Peer review of "Dietary Intake Influences Metabolites in Healthy Infants: A Scoping Review"

_nutrients, 2020, doi:10.3390/nu12072073_

Round 1
Reviewer 1 Report
The scoping review on metabolites profile in infants fed either breastmilk or formula diet is timely. Authors have done a good job in gathering the literature and summarizing different studies in table format. The following edits would provide interest to the reader.
- How did authors come to the conclusion tat fecal material is likely better than other sample types for metabolite profile? No discussion about common observations in three sample types or what differences observed among the sample types.
- It feels like a list of studies, does not interpret what the responses could mean for neonates health, even speculation since this is a review.
- What does the specific metabolite profile of breast-fed versus formula fed mean in terms of dietary differences in infants? There are differences in breastmilk versus formula milk composition, interpretations about the changes observed related to those would be appealing to the reader.
- Are there bacterial metabolites observed and how does that differ in different studies?
Reviewer 2 Report
This research involved a scoping review of the literature regarding metabolites in biofluids by infant diet to (1) assess the overall quality of the research; and (2) identify metabolites that were discriminatory between breastfed and formula fed infants. Novel dietary assessment tools using biomarkers has the potential to improve nutrition research through objective versus subjective data collection. Biomarker research can also provide insights on mechanistic pathways contributing to health differences by feeding type, thus this is a relevant and interesting research topic. There are several areas in this manuscript that need revision to improve clarity and stay within scope. There are also formatting issues with the tables.
Line 92-93. Why were studies that analyzed a single or handful of metabolites using ELISA excluded? This is not in agreement with your methods where you indicated that studies were included “regardless of technique used to analyze the metabolites” (line 77-78).
Figure 1. It would be helpful to see the reason full-text articles were excluded so this figure can stand on its own.
Methods. Please explicitly state what data was abstracted from included studies (e.g. the data that you present in your tables). You state this in the results (line 154-157), but this belongs in the methods.
Line 125. Should read “measurement techniques” not “measures” (you don’t measure HPLC, NMR, these are techniques used to measure specific substances).
Line 162-166. These lines are confusing. Why is the sentence “methods for analyzing fecal metabolites have been reviewed (61)” included in your results? This is not one of the studies included in your review so this information seems like it should be part of the discussion or intro, not the results. Perhaps you are saying that all studies used analytical techniques that have previously been validated for analyzing fecal metabolites? This section should be revised to focus specifically on results from this study. The rest of this paragraph is written in a way that it is not clear if you are describing your results or stating previously agreed upon facts. Perhaps clarify language by saying, “Reviewed studies identified infant age as an important covariate for metabolomic analysis of fecal samples…”
Line 168. How many of the 7 studies were able to discriminate between any or no human milk exposure? This should be explicitly stated in the results to facilitate reader understanding. General note when presenting your results: the studies were already conducted so it would be more appropriate to discuss in the past tense. For example, instead of “fecal metabolites CAN discriminate between any or no exposure to human milk”, it would be clearer and more specific to say “6/7 studies WERE able to discriminate between any or no human milk exposure using fecal metabolites.”
Table 1. Data don’t align under columns from “Diet” all the way to the right, making this table difficult to read. Same issue with the other tables.
Overall Results. You should also provide information in your results regarding how many studies reported a discriminatory capability. Right now, that information is only presented in the abstract, but it needs to be in the main body of the paper, too (the abstract should be a short summary of key points presented in the paper, not new information).
Line 177-181. Starting with the sentence “Taken together it is known…” is not results, but interpretation, and belongs in the discussion, not the results. Please simply describe a summary of what you found in the study (e.g. “Plasma metabolites included X, Y, and Z. 4/5 studies using plasma reported discriminatory results.”).
Line 193. Citation 63 is not one of your included studies and should not be reported in the results section. Similar to the other tables, it would be helpful if your results provide the reader with a narrative summary of the results in Table 3 (# of studies, main metabolites assessed, main methods of assessment, # studies reporting discrimination, etc.), and save interpretation and discussion of other literature for the discussion.
Line 206. You have not supported the comment “provides evidence that metabolites in fecal samples may be best suited for differentiating…” because you didn’t discuss this in results (only in the abstract).
Line 210. You have not supported the comment “measured using a variety of methods” as that wasn’t discussed in the results. Suggest including this summary in the results section of the manuscript (it is in the table, but you leave the reader to interpret the table when you don’t summarize the information for them).
Line 211. Need to define targeted and untargeted approach somewhere (either in intro or discussion).
Discussion: The discussion should circle back to your original objectives which were diet to (1) assess the overall quality of the research; and (2) identify metabolites that were discriminatory between breastfed and formula fed infants. The current discussion partial addresses #2 (identifies fecal biospecimen, but doesn’t discuss the metabolites that are discriminatory) and doesn’t address objective 1.
Conclusion. Your objective was not to describe gaps in the literature, so this conclusion feels disconnected from the objective of your review. Both the Discussion and Conclusion should be revised with the overall objectives of this research in mind.
Round 2
Reviewer 2 Report
The authors have done a nice job revising the manuscript. I had a few minor suggestions.
I am still confused why studies that targeted 3 or fewer metabolites were excluded compared to studies that targeted more metabolites. Could the authors provide a sentence with some additional explanation for this exclusion?
Your manuscript seems to still contain some notes between authors (e.g. lines 200 to 220 have several comments that read “please cite PMID XXX” as do lines 233 to 255).
Line 251 to 253. This sentence is confusing. Sounds like 175 infants were all exclusively fed human milk in this study, but then it mentions no difference between groups. Please clarify what the groups were.
